# Cytoplasmic p53β Isoforms Are Associated with Worse Disease-Free Survival in Breast Cancer

**DOI:** 10.3390/ijms23126670

**Published:** 2022-06-15

**Authors:** Luiza Steffens Reinhardt, Kira Groen, Brianna C. Morten, Jean-Christophe Bourdon, Kelly A. Avery-Kiejda

**Affiliations:** 1School of Biomedical Sciences and Pharmacy, College of Health, Medicine and Wellbeing, The University of Newcastle, Newcastle, NSW 2305, Australia; luiza.steffensreinhardt@uon.edu.au (L.S.R.); kira.groen@newcastle.edu.au (K.G.); brianna.morten@uon.edu.au (B.C.M.); 2Hunter Medical Research Institute, Newcastle, NSW 2305, Australia; 3School of Medicine, Ninewells Hospital and Medical School, The University of Dundee, Dundee DD1 9SY, UK; j.bourdon@dundee.ac.uk

**Keywords:** breast cancer, p53 isoforms, *TP53* mutation status, disease-free survival, Δ133p53β

## Abstract

*TP53* mutations are associated with tumour progression, resistance to therapy and poor prognosis. However, in breast cancer, *TP53*′s overall mutation frequency is lower than expected (~25%), suggesting that other mechanisms may be responsible for the disruption of this critical tumour suppressor. p53 isoforms are known to enhance or disrupt p53 pathway activity in cell- and context-specific manners. Our previous study revealed that p53 isoform mRNA expression correlates with clinicopathological features and survival in breast cancer and may account for the dysregulation of the p53 pathway in the absence of *TP53* mutations. Hence, in this study, the protein expression of p53 isoforms, transactivation domain p53 (TAp53), p53β, Δ40p53, Δ133p53 and Δ160p53 was analysed using immunohistochemistry in a cohort of invasive ductal carcinomas (*n* = 108). p53 isoforms presented distinct cellular localisation, with some isoforms being expressed in tumour cells and others in infiltrating immune cells. Moreover, high levels of p53β, most likely to be N-terminally truncated β variants, were significantly associated with worse disease-free survival, especially in tumours with wild-type *TP53*. To the best of our knowledge, this is the first study that analysed the endogenous protein levels of p53 isoforms in a breast cancer cohort. Our findings suggest that p53β may be a useful prognostic marker.

## 1. Introduction

Breast cancer is the most common malignancy in women. In 2020, approximately 2.3 million women were diagnosed with breast cancer, and 685,000 women died as a result of this disease [1]. *TP53* is a tumour suppressor, the somatic mutation of which in breast cancer is associated with tumour progression, resistance to therapy and poor prognosis. However, its overall mutation frequency of approximately 25% in breast cancer [2,3] is less than the mean percentage of *TP53* mutations in cancer (approximately 50%) [2], suggesting that other mechanisms may be responsible for the disruption of this critical tumour suppressor and its pathway in breast cancer.

*TP53* is expressed as full-length p53 (FLp53, p53α or wild-type p53) as well as 11 smaller isoforms: p53β, p53γ, Δ40p53 (α, β, γ), Δ133p53 (α, β, γ) and Δ160p53 (α, β, γ) [4,5,6], generated through alternative splicing, alternative promoter usage, the alternative initiation of translation or the post-translational degradation of p53 via the 20S proteasome [4,5,7,8,9,10,11]. Previous studies have shown that p53 isoform mRNA expression is dysregulated in several cancers [4,12,13,14,15,16,17,18,19], where the p53 isoforms have been associated with prognosis [14,18] and chemotherapy response [12,15]. However, a detailed understanding of their contribution to breast cancer prognosis is still lacking. Our previous study revealed the clinical relevance of p53 isoform expression in breast cancer [20], showing that of the p53 isoforms, Δ40p53 is the most highly expressed isoform at the mRNA level, and that Δ40p53 expression is significantly upregulated in tumours and cancer cell lines compared to normal adjacent breast tissues. Moreover, a high Δ40p53:p53 ratio (>0.7) was significantly associated with worse disease-free survival [20,21]. p53β, the second most highly expressed p53 isoform, was associated with increased disease-free survival, particularly in patients with mutated *TP53* [20].

Given the known roles of p53 isoforms in regulating the functionality of the wild-type protein, both negatively and positively depending on the context, with Δ40p53 and p53β capable of modulating gene expression and defining cell responses to cell signals in a p53-dependent and independent manner, our previous results have clear implications for the ability of *TP53* to function as a tumour suppressor in breast cancer. p53 isoforms are regulated post-transcriptionally [4,10,11,22,23,24,25], highlighting the need to validate our RNA findings at the protein level. Therefore, this study aimed to verify the mRNA expression of p53 isoforms in breast cancers with known *TP53* mutation status at the protein level with immunohistochemistry (IHC) using a suite of specific antibodies.

## 2. Results

### 2.1. Characterisation of p53 Isoform Localisation in Breast Cancer Tissues

To determine whether the endogenous expression of p53 isoforms at the mRNA level [20] could be verified at the protein level, we examined the same cohort of breast cancers using IHC (patient information and clinical diagnoses are detailed in Table 1). Initially, transiently transfected or stably transduced (for Δ40p53 only [26]) MCF-7 breast cancer cells were used as positive controls for the optimisation of antibody titrations for TAp53, and the p53β, Δ40p53, Δ133p53 and Δ160p53 isoforms for use in IHC (Figure 1). All p53 isoforms were detected in breast cancer cells, and their subcellular localisation differed from isoform to isoform (Figure 1B). Interestingly, all isoforms, including the ones known to be also localised in the cytoplasm such as Δ40p53 [17,27], were mainly localised in the nucleus, indicating that the overexpression of the isoforms may induce the nuclear upregulation or translocation of p53 isoforms. This finding suggests an alteration of the cellular functions of the isoforms when overexpressed, which may not correlate with the isoforms’ activities at the basal level under normal conditions.

By using optimised IHC conditions for each of the antibodies, 108 breast cancer specimens comprising 31 Grade 1, 24 Grade 2 and 53 Grade 3 invasive ductal carcinomas were immunostained, and p53 isoform expression patterns were investigated (Figure 2A). TAp53 was predominantly localised within the tumour cells, whereas p53β isoforms and Δ160p53 (Δ160p53α isoform) were localised within tumour cells and tumour-infiltrating lymphocytes. In contrast, Δ40p53 was found predominantly within tumour-associated-plasma cells, and Δ133p53 (Δ133p53α isoform) was also found in plasma cells and tumour cells (Figure 2A).

As expected, the isoforms presented distinct subcellular localisation (Figure 2A); TAp53 was found predominantly in the nucleus, whereas the β isoforms, Δ40p53, Δ133p53 and Δ160p53 were found in both the nucleus and cytoplasm.

### 2.2. Characterisation of p53 Isoform Expression in Breast Cancer Tissues

The p53 isoform expression varied considerably among specimens and within the same tissue, with some tissues showing positive cells for all analysed isoforms and some samples presenting strong staining for nuclear TAp53 but weak or negative staining for all other isoforms tested (Appendix A). TAp53 staining varied from weak to strong, and the staining was exclusively strong for the nuclear expression. For Δ40p53, the staining was strong for the few cells expressing the isoform in the cytoplasm and weak for the nuclear expression. p53β, Δ133p53 and Δ160p53 staining was predominantly categorised as weak, with some samples presenting moderate staining. As expected, TAp53 presented the highest H-score (*p* < 0.0001 when compared to Δ40p53) [20] (Figure 2B). Interestingly, TAp53 staining was highly variable, with the percentage of positive cells (%) ranging from 0.23 to 93.95% of cells for the nuclear staining and 0 to 92.86% for the cytoplasmic staining (Figure 2C).

p53β, Δ40p53, Δ133p53 and Δ160p53 presented a higher median percentage of positive cells (%) for the nuclear staining when compared to the cytoplasmic staining (*p* < 0.0001) (Figure 2C). The percentage of nuclear p53β-, Δ133p53- and Δ160p53-positive cells ranged from 0% to 76.91%, 0.29 to 82.65% and 0.69 to 47.94%, respectively, and cytoplasmic p53β, Δ133p53 and Δ160p53-positive cells ranged from 0% to 46.90%, 0 to 33.38%, and 0 to 28.28%, respectively, across patient samples (Figure 2C). Less than 14% of cells were positive for Δ40p53 (range of 0–1.30% for cytoplasmic staining and 0–13.80% for nuclear staining) (Figure 2C).

### 2.3. The association of p53 Isoform Expression with Tp53 Mutation Status and Clinicopathological Features

To determine if p53 isoform expression was related to the mutation status of p53, *TP53* mutation data from our previous study were used [20]. Truncating mutations were present in 28% of the cohort and were associated with age, grade, tumour size, oestrogen receptor (ER), progesterone receptor (PR), human epidermal growth factor receptor 2 (HER2) and triple-negative breast cancer (TNBC) classification (Table 2). We analysed whether the percentage of positive cells for each p53 isoform was associated with deleterious mutations in *TP53* and clinical features using multiple linear regression (Appendix A). Δ40p53 was excluded from the analysis since its expression was practically negative within tumour cells. With the exception of TAp53, with which nuclear staining was positively associated with *TP53* mutation status (*p* = 0.0022) and cytoplasmic staining was negatively associated with tumour size (*p* = 0.0329), the expression of the isoforms was not associated with the mutation status of *TP53* or clinical features. TAp53 positivity was significantly higher in ER− (vs. ER+ *p* < 0.01) and PR− (vs. PR+ *p* < 0.01) specimens, and it was higher in TNBC (ER−PR−HER2−) specimens when compared to ER+PR+HER2− (*p* < 0.05) and ER+PR−HER2− (*p* < 0.01) subtypes (Figure 3A). No statistically significant differences were found for the other isoforms (Figure 3B–D).

In breast cancer, high levels of p53 have been found to be associated with worse prognosis [28] and *TP53* mutation status since wild-type p53 is less stable than mutated p53 [29]; however, it remains unclear whether p53 positivity has predictive significance [29]. Thus, TAp53 nuclear expression was segregated into low and high levels (by using the median expression as a cut-off value), and its relationship with clinical features was evaluated. Amid the analysed features, tumour size (*p* = 0.0039) was significantly negatively associated with TAp53 expression, whereas age at diagnosis (*p* = 0.0361) and TNBC specimens (*p* = 0.0123) were significantly positively associated with TAp53 expression (Table 3). The same analysis was performed with all other isoforms; however, no statistically significant differences were found (data not shown).

Next, we analysed the correlation between TAp53 levels and p53 isoform expression and between the different p53 isoforms. Interestingly, except for nuclear Δ133p53 and Δ160p53, the expression of which correlated with TAp53 expression at low levels (Figure 4A), the other p53 isoforms did not correlate with TAp53 (Figure 4). This result indicates that p53 isoform expression may be regulated differently than the full-length protein in breast cancer tissue. Moreover, Spearman rank analysis showed that the expression of Δ133p53 and Δ160p53 and the expression of p53β and Δ133p53 were correlated (*p* = 0.002), but no other correlations were found (Appendix A).

### 2.4. p53 Isoform Expression Association with Disease-Free Survival

Next, we evaluated whether disease-free survival (time to metastasis or recurrence) was associated with *TP53* mutation status or p53 isoform expression. No association was found between disease-free survival and mutated *TP53* versus wild-type (Figure 5A), or when cases expressing negative or strong TAp53 [30] were compared (Figure 5B); however, we observed a trend of less metastases in specimens expressing moderate TAp53, suggesting that negative or strong TAp53 could contribute to worse disease-free survival.

No statistically significant differences were found in disease-free survival when cases expressing high levels (using the median expression as the cut-off between high and low) of nuclear p53β (Figure 5C), Δ133p53 or Δ160p53 (data not shown) were compared with those expressing low levels of these isoforms.

In contrast, disease-free survival was significantly different in cases expressing high levels of the cytoplasmic p53β isoform compared to those expressing low levels (Figure 5D). Interestingly, disease-free survival was significantly different when cases expressing high levels of cytoplasmic p53β with wild-type *TP53* were compared with those expressing high or low levels of cytoplasmic p53β with truncated *TP53* or cases expressing low levels of cytoplasmic p53β with wild-type *TP53* (Figure 5E), suggesting that high levels of cytoplasmic p53β isoforms contribute to a worse prognosis independently of *TP53* mutations.

Given that Δ133p53 and p53β expression were correlated and that p53β was not correlated with TAp53, next, we aimed to elucidate which p53β isoform was driving differences in disease-free survival: the TA intact isoform or an N-terminally truncated isoform such as the Δ133 isoform. In order to do this, dual-staining was performed on samples of nine patients who had a recurrence and/or metastasis using the DO-1 antibody (which recognises p53α, p53β and p53γ), the KJC8 antibody (which recognises p53β variants) and two different chromogens (DAB for DO-1 and Fast Red for KJC8). Thus, cells that stained for both chromogens were most likely p53β, whereas cells stained in red were most likely Δ133p53β or Δ160p53β (since Δ40p53 expression was practically negative in tumour cells: Figure 2). Cells only stained in brown were most likely p53α (Figure 6A). Within these specimens, an increase in the number of KJC8-positive cells was observed when compared to DO-1-positive cells, which presented negative or weak staining (Figure 6B). Moreover, cells expressing p53β isoforms in the cytoplasm presented foci-like staining, as previously reported for Δ133p53β [4,5] (Figure 6A, B). Cells presented a higher expression of TAp53 in the nucleus (DO-1 n) and p53β isoforms in the cytoplasm (KJC8 c) (Figure 6C–E). Some cells were positive for both chromogens when TAp53 was nuclear (KJC8 and DO-1 n) (Figure 6D,E); however, only a small percentage of cells (<1%) was positive for both chromogens when TAp53 was cytoplasmic (KJC8 and DO-1 c) (Figure 6D,E). These findings indicate that in these specimens, the expression of the TA intact p53β isoform is low and that increased levels of N-terminally truncated β variants may play a role in the worse prognosis of these cases.

### 2.5. p53 Isoform mRNA Levels Do Not Correlate with Their Protein Product in Breast Cancer

The aforementioned associations between p53 isoform expression and clinical parameters or disease-free survival were not expected, since our previous mRNA studies showed disparate results [20,21]. Therefore, the correlation between the relative mRNA expression of the p53 isoforms (p53α, p53β, Δ40p53 and Δ133p53) [20] and the H-scores was evaluated (Figure 7A). No correlation was found for any isoform, possibly explaining why the results found here were different from our previous study [20].

In our previous study, Δ40p53 was found to be the highest-expressed p53 isoform (when compared to the p53β, p53γ and Δ133p53 isoforms) at the mRNA level [20], but in the current study, it was the lowest expressed at the protein level (Figure 2) and was found predominantly in plasma cells (Figure 2). Considering that when performing RNA extractions there is no separation of tumour cells and other cells found in the tumour microenvironment, the in situ hybridisation of two TNBC samples that contained high mRNA levels of Δ40p53 was performed to confirm the localisation of Δ40p53 mRNA (Figure 7B, Appendix A). The results clearly showed that Δ40p53 mRNA was present in the tumour cells similarly to the full-length variant but at lower levels (Figure 7B, Appendix A). Hence, as suggested by this study, Δ40p53 mRNA levels are detectable in breast cancer tissue; however, its protein product is expressed at low levels.

## 3. Discussion

p53 isoforms play an essential role in cell fate decisions through their regulation of p53 tumour suppressor functions. However, there have been limited studies performed on endogenously expressed isoforms at the protein level owing to a lack of specific antibodies. In this study, we used a suite of antibodies to characterise p53 isoform expression in breast cancer tissues. We showed that the p53 isoforms were expressed at the protein level in distinct cellular and subcellular localisations with a highly variable expression pattern among breast cancer tissues.

Possibly the most important finding from this study, high levels of cytoplasmic p53β isoforms were significantly associated with worse disease-free survival when compared to specimens expressing low levels (Figure 5D), especially in tumours with wild-type *TP53* (Figure 5E). These results challenge our previous findings [20] and a recent investigation [31]. Thus, by performing dual-staining, we hypothesised that N-terminally truncated β variants such as Δ133p53β, not TAp53β levels, may be driving the association with worse disease-free survival found here (Figure 6). In agreement with this, several studies have described high levels of Δ133p53β associated with worse prognosis and survival outcomes in different cancers [32,33,34,35,36]. Δ133p53β has been associated with cell invasiveness and cancer recurrence, particularly in luminal A breast cancers with wild-type *TP53* [33]. This isoform may promote a stemness phenotype by stimulating the expression of pluripotency markers such as *SOX2*, *NANOG* and *OCT3/4* [37] and may act as an oncogene by inhibiting chemotherapy-induced apoptosis by directly binding to anti-apoptotic small GTPase, RhoB [34]. Additionally, Δ133p53β may compete with p53 to bind p53-response elements on DNA [38], driving altered transcriptional regulation, leading to the enrichment of genes involved in immune regulation, including interferon-γ (the upregulation of *JAK2*, *STAT6* and *IL6ST*), PD-1 signalling (the upregulation of *CD274*) and cell invasion (the activation of matrix metalloproteinases) [36]. Similar to mutated p53, Δ133p53β forms protein aggregates (equivalent to the foci-like staining found in this study, shown in Figure 6), and when the isoform was recruited, cancer cell invasion was activated [39]. These investigations provide a mechanistic hypothesis for an association between Δ133p53β and worse prognosis, since endogenous Δ160p53β has not been described and may be unstable [38]. Hence, Δ133p53β analysis in breast tumours may help clarify the discrepancies between *TP53* mutation status and prognosis in the literature; moreover, Δ133p53β could be used as a prognostic biomarker.

Significant variability was observed in the subcellular localisation of the isoforms and in the overall p53 isoform expression from specimen to specimen (Figure 2). TAp53 (most likely to predominantly represent the full-length isoform since its expression was stronger when compared to any other p53 isoform staining) was primarily found in the nucleus, whereas the other isoforms were found in both cellular compartments. It has been described that p53β is mainly localised in the nucleus [4]; however, Δ133p53β is localised in the nucleus and the cytoplasm, and as previously mentioned, it may exhibit cytoplasmic foci-like staining [4,5]. This may explain why p53β-positive staining was found in both cellular compartments in this study. Both Δ133p53α and Δ160p53α have been described to localise in the nucleus with minor staining in the cytoplasm; however, Δ160p53α can also present peri nucleolar localisation [4,5,40]. This indicates that these isoforms may be travelling between the cytoplasm and the nucleus; however, isoform translocation within cellular compartments could depend on the cellular context (e.g., in response to DNA damage).

Regarding the differential expression of the isoforms, our findings are consistent with studies showing that the p53 isoform expression differs from tumour to tumour, with some truncated *TP53* breast cancers expressing wild-type p53 isoforms [4], and that p53 isoform expression may not be associated with *TP53* mutation status or clinicopathological parameters [18]. Moreover, p53 isoform expression may fluctuate in different cell subpopulations depending on their proliferation status [16].

The failure to regulate p53 isoform levels could contribute to tumour development, therapy outcomes and prognosis [41]. Given that only 28% of the breast cancers analysed in this work harboured truncating mutations in *TP53*, we assumed that other mechanisms not associated with *TP53* mutation status, including p53 isoform expression, may affect breast cancer outcomes via the p53 pathway. In this study, *TP53* mutation status was not associated with disease-free survival (Figure 5A). On the other hand, both negative and strong TAp53 H-scores revealed trends towards worse disease-free survival (Figure 5B), while patients with moderate levels experienced less recurrence and/or metastasis. This result is in agreement with a larger breast cancer cohort investigation (*n* =293) [30], and strongly suggests that the regulation of p53 levels affect patients’ prognosis and that appropriate levels of p53 are essential for maintaining genomic stability.

In this study, the results associated with p53β and Δ40p53 isoforms did not support our previous findings at the mRNA level, where p53β expression was associated with increased disease-free survival and Δ40p53 expression was associated with the TNBC subtype [20]. The lack of correlation between protein and mRNA levels (Figure 7A) offers a possible explanation for this discrepancy. Several studies have shown that the correlation between mRNA expression and protein products is poor (~40%) [42,43,44], with some classes of genes presenting higher correlations, including differentially expressed mRNAs when comparing conditions (e.g., treated versus non-treated) [45]. The discrepancy between p53 isoform protein and mRNA levels could be attributed to the transcription regulation program of p53, which is extensively regulated by several factors and upstream signalling [25,46,47,48,49,50]. Moreover, p53 isoforms can be regulated post-translationally [11], which could further modulate protein levels and account for the results observed here.

The lack of correlation between mRNA and protein levels may create concerns about the biological relevance of p53 isoform mRNA analyses. However, for p53 isoforms generated through alternative splicing, such as the C-terminally truncated variant p53β, the mRNA and protein levels may be equivalent, given that it is solely produced via the alternative splicing of mRNA. Thus, mRNA analyses could provide accurate values when compared to IHC scores since the KJC8 antibody recognises different β variants. Contrastingly, for truncated isoforms that are generated via alternative translation or post-translational mechanisms such as Δ40p53, which was the lowest expressed isoform at the protein level (Figure 2), but is highly expressed in several tumour tissues at the mRNA level (Figure 7) [12,16,17,18,19,20], protein level measurement seems to be more appropriate than mRNA quantification. While some studies confirmed Δ40p53 expression at the protein level [16,18,20], the majority of studies used mRNA data to perform association analyses. Δ40p53 protein-level evaluation may still be pertinent since this isoform has important functions in cell fate decisions following treatment with genotoxic agents [12,22,27,51,52,53]. In addition, this study evaluated therapy-naïve samples, but Δ40p53 expression may be modulated following standard cancer treatment. Thus, it is possible that Δ40p53 may still serve as a prognostic tool to indicate the response to DNA-damaging therapies in breast cancer.

Even though our p53 isoform mRNA associations were not supported by this study, the IHC analysis of the p53 isoforms highlighted important features regarding p53 isoform subcellular and cellular localisation, suggesting possible important functions for these variants in regard to immune cell regulation and the tumour microenvironment. It is known that p53 can regulate immune signalling pathways by transactivating genes that play key functions in these pathways [54,55]. With increased levels of p53β, Δ40p53, Δ133p53, and Δ160p53 detected in immune cells (Figure 2A), it seems that the isoforms may drive some of p53′s functions. Previous investigations have shown associations between p53 isoform expression and the regulation of immune cells [19,56,57,58], underscoring significant roles including the regulation of senescence in T lymphocytes [56,57,58] and improved treatment response in tumour-bearing mice [56]. Nevertheless, given that p53 isoform expression is cell- and isoform-dependent, the functional role of high levels of p53 isoforms in immune cells surrounding the breast tissue warrants further investigation.

To the best of our knowledge, this is the first study reporting high levels of N-terminally truncated p53 isoforms, Δ40p53 and Δ133p53 in tumour-infiltrating plasma cells (Figure 2A). These cells are B cells that differentiate into antibody-secreting cells after antigen exposure, and the findings in the literature have shown contradictory conclusions regarding the role of plasma cells in breast cancers [59,60,61,62,63,64]. Δ40p53 has not been previously described as associated with immune-related signalling pathways; however, the role of Δ133p53 has been partially investigated [65,66]. Δ133p53 may drive autoimmunity and chronic inflammation phenotypes via high pro-inflammatory cytokine and transcript expression and augmented levels of auto-antibodies often identified in systemic autoimmune disorders (reviewed in [65]). Therefore, Δ133p53 expression could contribute to immune cell infiltration in breast tumours given that inflammation signalling plays a major role in cancer.

## 4. Materials and Methods

### 4.1. Subcloning of p53 Isoform Plasmid Vectors

Plasmid vectors for full-length wild-type p53 (pRcCMV-p53WT) were generously provided by Prof. Helen Rizos (Department of Biomedical Sciences, Macquarie University, Sydney, Australia). The constructs for p53β (pSV40-p53β), Δ133p53 (pSV40133-p53), Δ160p53 (pCDNA3.1-Δ160p53α) and the control vector pSV40 were kindly provided by Dr. Jean-Christophe Bourdon (Cellular Medicine, School of Medicine, The University of Dundee, Dundee, Scotland, UK) and have been previously described [4,12,40,52].

### 4.2. Transfection of p53 Isoforms Plasmids into Breast Cancer Cells and Cell Pellet Preparation

The human breast adenocarcinoma cell line MCF-7 was kindly provided by Professor Christine Clarke (Westmead Millennium Institute, The University of Sydney, Sydney, Australia) and maintained in DMEM (Dulbecco modified Eagle’s medium) supplemented with foetal bovine serum (10% *v/v*), insulin (10 µg/mL) and L-glutamine (2 mM) (Life Technologies, Mulgrave, VIC, Australia) in humidified 5% CO_2_ at 37 °C. The cells were seeded at a density of 3 × 10^4^ per well in a 6-well plate 24 h before transfection. Cells were transfected in triplicate with 500 ng/well of the plasmid vectors using Lipofectamine 2000 (Life Technologies, Mulgrave, VIC, Australia) according to the manufacturer’s recommendations. Following 24 h of transfection, the cells were harvested for cell block preparation according to the method detailed in [67]. Briefly, cells were transferred to a 1.5 mL microcentrifuge tube, centrifuged at 200× *g* for 5 min, fixed with 3.7% formaldehyde for 10 min, washed with PBS and re-centrifuged. The cell pellet was resuspended in 2% (*w/v*) ultra-low melting agarose, re-centrifuged, and the mixture was let to solidify at 4 °C overnight. The solidified agarose pellet was removed into a cap of a microcentrifuge tube and embedded in 2% (*w/v*) normal agarose. Once solidified, the gel disk was removed from the cap using a needle and placed in a tissue cassette for paraffinisation as part of routine tissue processing. For Δ40p53-expressing cell pellets, previously generated MCF-7 cells overexpressing Δ40p53 were used [26]. The transfected cell blocks were used as IHC-positive controls for individual isoforms (Figure 1B).

### 4.3. Study Cohort

One hundred and eight formalin-fixed, paraffin-embedded (FFPE) breast tumour slides cut into 4 µm sections and mounted onto coated slides were kindly provided by the Australian Breast Cancer Tissue Bank (Westmead, NSW, Australia). The samples included 31 Grade 1, 24 Grade 2 and 53 Grade 3 invasive ductal carcinomas. Patient information and clinical diagnoses are detailed in Table 1. This was conducted in accordance with the Helsinki Declaration with ethical approval from the Hunter New England Human Research Ethics Committee (approval number: H-2009-0265). All patients agreed to the use of their clinical information and tissue in this study.

### 4.4. Antibodies

The mouse monoclonal antibody DO-1 (detects the transactivation domain p53 isoform, TAp53: p53α, p53β and p53γ) [4], and rabbit polyclonal antibodies KJC8 (detects the p53β isoforms) [4,35], KJC40 (detects the Δ40p53 isoforms, mainly Δ40p53α) [26], KJC133 (detects the Δ133p53α isoform) and KJC160 (detects the Δ160p53α isoform) (Figure 1A) were purchased from the University of Dundee (antibodies were developed by Dr. Jean-Christophe Bourdon, Dundee, Scotland, UK).

### 4.5. Immunohistochemistry

The IHC, including deparaffinisation to staining with diaminobenzidine (DAB) substrate, was performed by the NSW Regional Biospecimen & Research Services (Newcastle, NSW, Australia) using the Ventana Discovery Automated Immunostainer (Roche Medical Systems, Tuscon, AZ, USA). Tissue sections were deparaffinised using EZPrep (Roche Medical Systems, Tuscon, AZ, USA) at 95 °C. Slides were incubated for 40 min in RiboCC (pH 6) or CC1 (pH 9) Ventana solutions for antigen retrieval at pH 6 (DO-1) and pH 9 (KJC8, KJC40, KJC133 and KJC160). After antigen retrieval, slides were incubated for 12 min with a peroxidase inhibitor (Roche Medical Systems, Tuscon, AZ, USA) to inactivate endogenous peroxidase activity. Next, tissue sections were incubated with primary antibodies at a dilution of 1:160 (DO-1), 1:40 (KJC8 and KJC40) or 1:100 (KJC133 and KJC160) for 32 min at 37 °C. The pre-diluted anti-mouse hapten (HQ) or anti-rabbit HQ secondary antibodies (Roche Medical Systems, Tuscon, AZ, USA) were then added, and slides were incubated for 16 min at 37 °C followed by incubation with anti-HQ-horseradish peroxidase (HRP) (Roche Medical Systems, Tuscon, AZ, USA) tertiary antibody for 16 min at 37 °C. For the KJC133 slides, amplification was carried out with a Discovery Amplifier (8 min of incubation at 37 °C). The immunolocalised isoforms were visualised using a DAB chromogen detection kit (Roche Medical Systems, Tuscon, AZ, USA). For the dual-staining of DO-1 and KJC8 antibodies, the procedure for DO-1 was performed on each slide followed by the protocol for KJC8, where slides were incubated with anti-HQ-alkaline phosphatase tertiary antibody for 16 min at 37 °C and were visualised using a Naphthol solution and a Fast Red chromogen detection kit (Roche Medical Systems, Tuscon, AZ, USA). All slides were manually counterstained with Mayers hematoxylin for 10 s and rinsed in Scott’s Tap Water for 30 s. Following this step, the slides were dehydrated through three changes of absolute alcohol (2 min each) and two changes of xylene (2 min each), and the slides were then sealed with a glass coverslip and allowed to dry. Slides were scanned at 40× magnification using an Aperio AT2 scanner (Leica, Wetzlar, Germany). Immunostained slides and the identification of cell types such as lymphocytes, tumour and plasma cells were histologically evaluated by an expert pathologist. Slides were analysed with HALO Software (Halo imaging analysis software, Indica Labs, Corrales, NM, USA) using the CytoNuclear v2.0.8 analysis mode, which automatically scores the staining intensity from weak to strong. On each image, five regions covering a tissue area of 204,000 µm^2^ were manually annotated. Annotations were selected to exclude tissue artefacts. H-scores and the percentage of positive cells [35] for each p53 isoform were quantified for each tumour.

### 4.6. In Situ Hybridisation

In order to confirm the Δ*40p53* and *TP53* mRNA expression, two tissue sections from cancers that were shown to highly express these genes at the mRNA level via real time-PCR were chosen and mounted onto positively charged glass slides. QuantiGene ViewRNA ISH (Life Technologies, Mulgrave, VIC, Australia) tissue assays were performed on the ThermoBrite hybridisation system (Leica Biosystems, North Ryde, NSW, Australia). Only the single-plex assay was performed to provide the greatest expression of the target RNA in the tissue sections. The Type 1 (Alexa Fluor 546) probe sets for Δ*40p53* (VA1-14024) [68], *TP53* (VA1-11152) [68] and *GAPDH* (VA1-10119) (Life Technologies, Mulgrave, VIC, Australia) were used. The hybridised tissues were stained with DAPI for cell nuclei visualisation, mounted onto coverslips using the Ultramount medium (Agilent Technologies, Santa Clara, CA, USA) and observed using confocal microscopy.

### 4.7. Statistical Analysis

All continuous variables were tested for normal distribution. The association between p53 isoform expression and clinical parameters was evaluated using multiple linear regression, as previously described [20]. The relationship between different levels of p53 isoform expression (low versus high) with clinical parameters was evaluated using Pearson’s chi-square test. The Kolmogorov–Smirnov test or Kruskal–Wallis test followed by Dunn’s multiple comparisons test were used to evaluate the relationship between p53 isoform expression and hormone receptor status and H-scores or the percentage of positive cells between isoforms. The relationship between p53 isoform expression and disease-free survival was performed using Kaplan–Meier analysis, as previously described [20,21]. Analyses of the *TP53* mutation status and clinical parameters, p53 isoform expression correlation with one another and the correlation between p53 levels and isoform expression or mRNA levels and H-scores were performed using Spearman’s correlation test. All statistical analyses were carried out in GraphPad Prism (Version 9) (GraphPad Software, La Jolla, CA, USA). An adjusted *p*-value of <0.05 was considered to be statistically significant.

## 5. Conclusions

Overall, this study demonstrated that high levels of cytoplasmic p53β, most likely to be an N-terminally truncated β variant such as Δ133p53β, were significantly associated with worse disease-free survival in invasive ductal carcinomas. Thus, p53′s, either wild-type or mutant, relationship with breast cancer metastasis and/or recurrence-associated events may be determined through the modulation of Δ133p53β levels. Moreover, the p53 isoforms presented distinct subcellular and cellular localisations, with some isoforms being expressed in immune cells, and others in tumour cells or both. Finally, to the best of our knowledge, this is the first study to investigate endogenous levels of p53 isoform protein expression in a breast cancer cohort, and as shown in this study, the use of a suite of specific antibodies may provide detailed analyses, making the investigation of p53 isoforms via IHC possible.

## Figures and Tables

**Figure 1 ijms-23-06670-f001:**
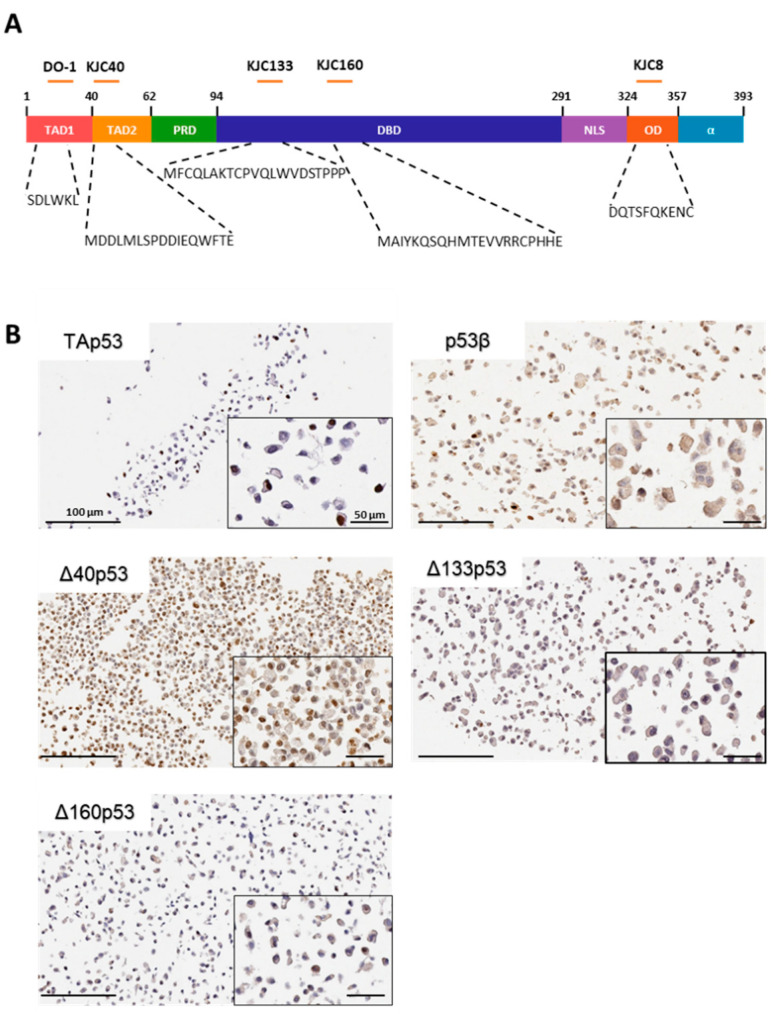
Characterisation of p53 isoform protein expression using immunohistochemistry. (**A**) Antibody epitopes within p53′s functional domains. TAD1: transactivation domain 1, TAD2: transactivation domain 2, PRD: proline-rich domain, DBD: DNA-binding domain, NLS: nuclear localisation signals, OD: oligomerisation domain and α: C-terminal domain. The mouse monoclonal antibody DO-1 detects the sequence SDLWKL of TAD1; the rabbit polyclonal antibody KJC40 recognises MDDLMLSPDDIEQWFTE with specific post-translational modifications; the rabbit polyclonal antibody KCJ8 recognises the region DQTSFQKENC of the alternatively spliced p53β sequence; the epitope for the rabbit polyclonal antibody KJC133 is between position 133 and 153, and the epitope for the rabbit polyclonal antibody KJC160 is between 160 and 180 using p53α amino-acid sequence. (**B**) Representative images of MCF-7 cells transfected with constructs for wild-type p53, p53β, Δ133p53α or Δ160p53α, or stably transduced for Δ40p53 and immunostained for TAp53 (DO-1), p53β (KJC8), Δ133p53 (KJC133), Δ160p53 (KJC160) or Δ40p53 (KJC40), respectively. Scale bar represents 100 and 50 µm for 20× and 40× images, respectively.

**Figure 2 ijms-23-06670-f002:**
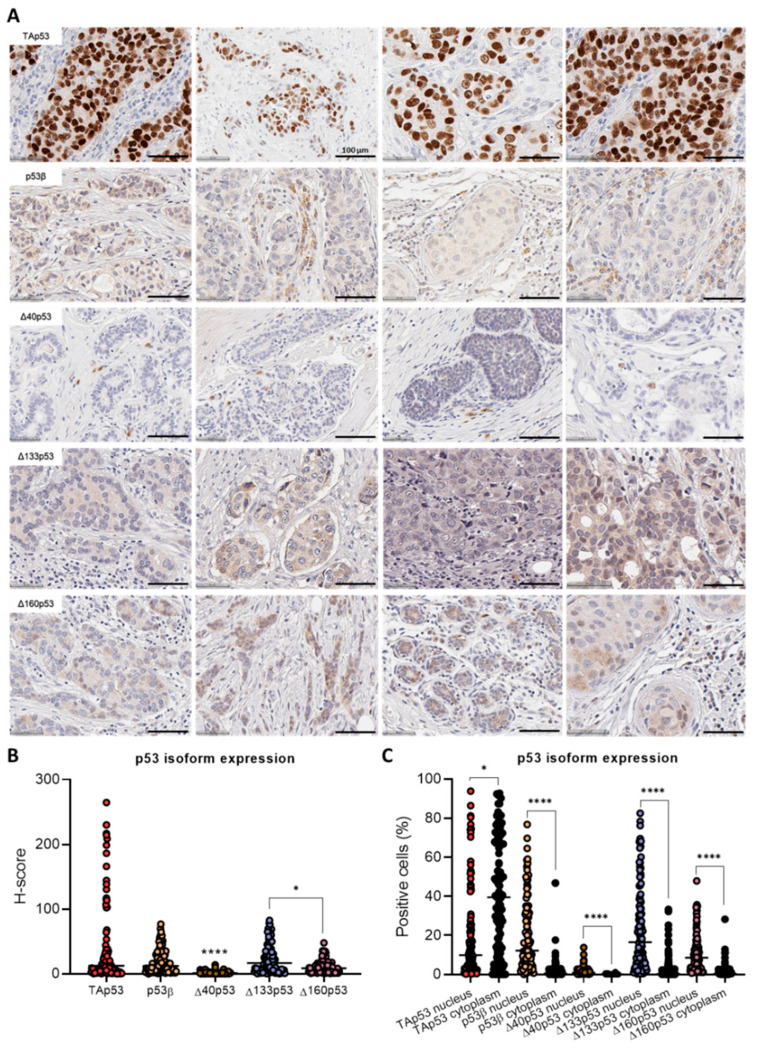
Immunohistochemistry analysis of p53 and its isoforms in 108 invasive breast cancer specimens. (**A**) Representative images of slides stained for TAp53 (DO-1), p53β (KJC8), Δ40p53 (KJC40), Δ133p53 (KJC133) or Δ160p53 (KJC160). Scale bar represents 50 µm unless otherwise specified. (**B**) H-score for each p53 isoform: TAp53 (DO-1), p53β (KJC8), Δ40p53 (KJC40), Δ133p53 (KJC133) or Δ160p53 (KJC160). (**C**) Percentage of cells positive for nuclear or cytoplasmic isoform expression. Results are shown as individual values, with the median represented by the black line. Statistical analyses were carried out using the Kruskal–Wallis test followed by Dunn’s multiple comparisons test. Results were considered significant at *p* < 0.05; * *p* < 0.05, **** *p* < 0.0001 when compared to TAp53 unless otherwise specified.

**Figure 3 ijms-23-06670-f003:**
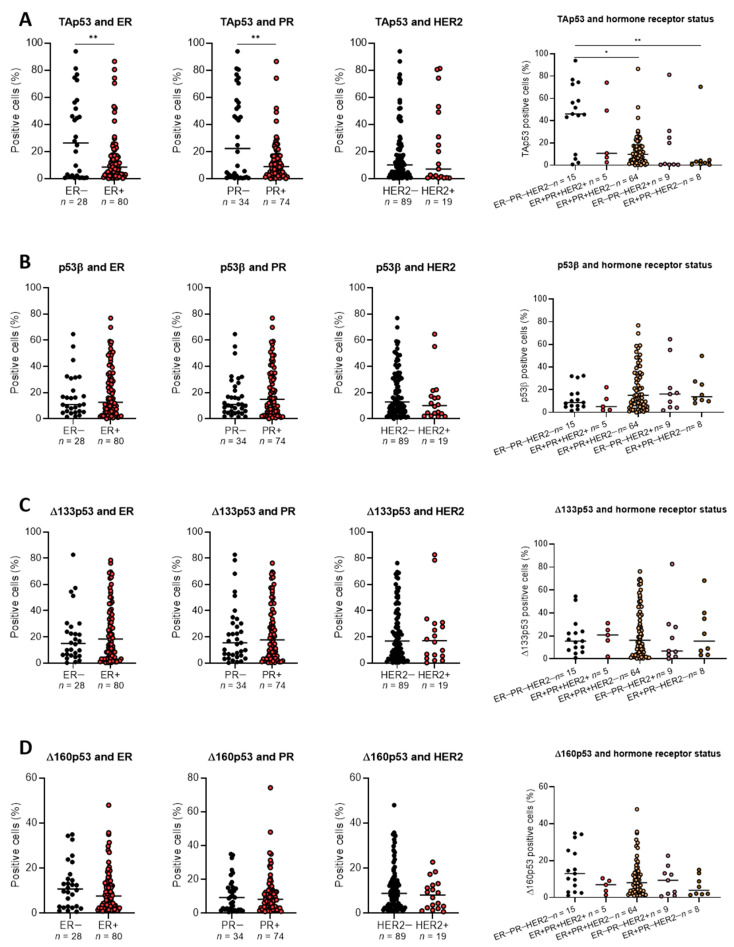
Association between hormone receptor status and p53 isoform expression. The samples (*n* = 108 or *n* = 107 for HER2-related analyses) were grouped based on their hormone receptor status (ER/PR/HER2), and the percentage of cells positive for each p53 isoform was analysed. (**A**) TAp53, (**B**) p53β, (**C**) Δ133p53, (**D**) Δ160p53. Results are shown as individual values with the median represented by the black line. Statistical analyses were carried out using Kolmogorov–Smirnov tests for positive or negative ER, PR or HER analyses. Kruskal–Wallis test followed by Dunn’s multiple comparisons test was used to determine the statistical significance of p53 isoform expression between the hormone receptor status subgroups. Results were considered significant at *p* < 0.05; * *p* < 0.05, ** *p* < 0.01.

**Figure 4 ijms-23-06670-f004:**
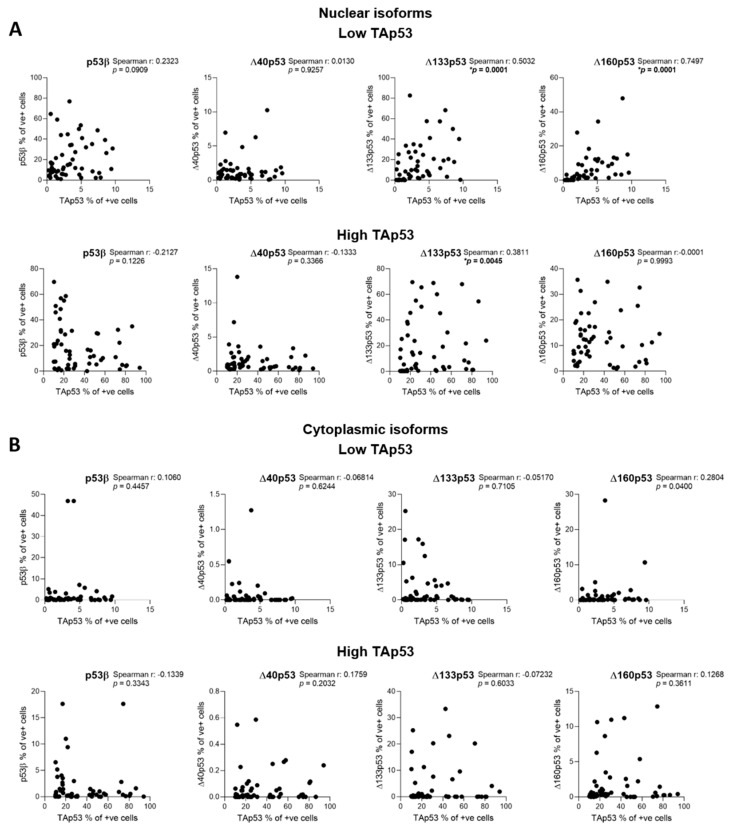
Correlation analyses between p53 isoform protein expression in samples with low (*n* = 54) and high (*n* = 54) TAp53 levels. The percentage of positive cells for (**A**) nuclear p53 isoforms or (**B**) cytoplasmic p53 isoforms was used. Spearman’s rank correlation test was used to assess the correlation between p53-isoform-positive cells and TAp53-positive cells. Results were considered significant at *p* < 0.05 (*).

**Figure 5 ijms-23-06670-f005:**
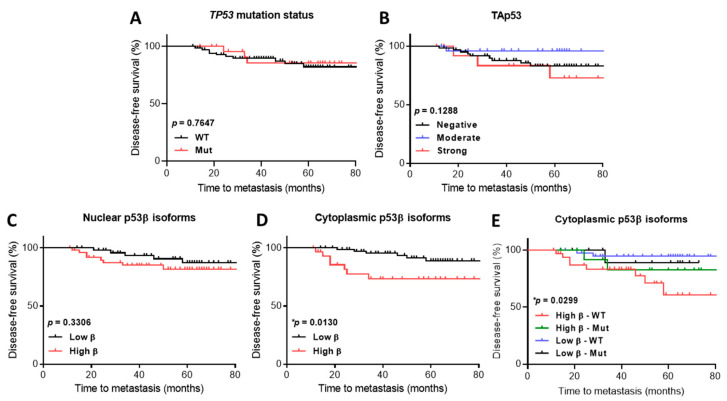
p53 isoform expression is associated with worse disease-free survival. Kaplan–Meier survival curve representing disease-free survival of (**A**) 98 patients who had *TP53* mutation status and metastasis data available [20]; cases were distributed based on *TP53* mutation status into wild-type (black, *n* = 72) or mutated (red, *n* = 26). (**B**) Cases (*n* = 103) segregated based on TAp53 expression into negative (black, *n* = 63), moderate (blue, *n* = 26) or strong (red, *n*= 14). (**C**) Cases (*n* = 103) segregated based on nuclear p53β expression into low (black, *n* = 53) or high (red, *n* = 50). (**D**) Cases (*n* = 103) segregated based on cytoplasmic p53β expression into low (black, *n* = 72) or high (red, *n* = 31). (**E**) Cases (*n* = 98) segregated based on cytoplasmic p53β expression and *TP53* mutation status into low p53β with mutated *TP53* (black, *n* = 12), low p53β with wild-type *TP53* (blue, *n* = 39), high p53β with mutated *TP53* (green, *n*= 14) or high p53β with wild-type *TP53* (red, *n*= 33). Log-rank (Mantel–Cox) test was used to determine statistical significance. Results were considered significant at *p* < 0.05 (*).

**Figure 6 ijms-23-06670-f006:**
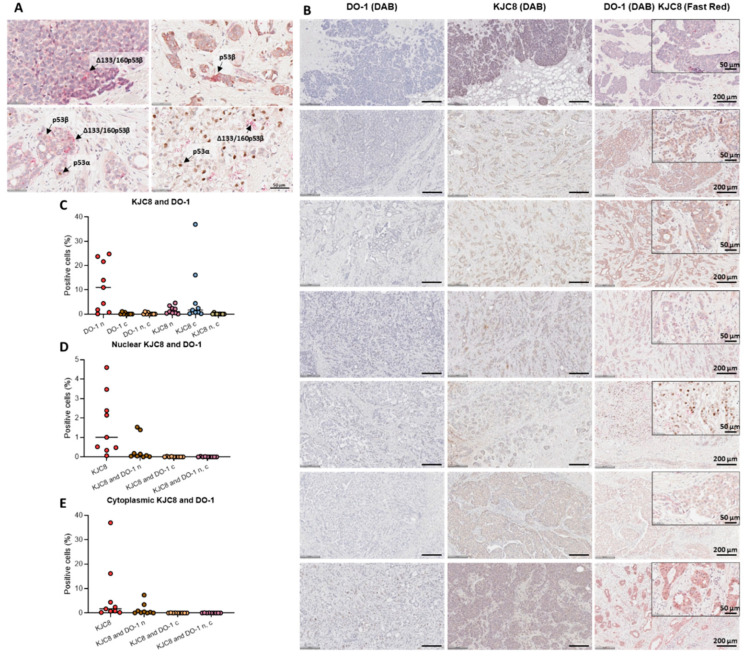
Dual-staining of DO-1 and KJC8 in nine breast cancers (identified as having a recurrence and/or metastasis). (**A**) Representative images of slides stained for TAp53 (DO-1, DAB) and p53β (KJC8, Fast Red). Scale bar represents 50 µm. (**B**) Representative images of slides stained for TAp53 (DO-1, DAB), p53β (KJC8, DAB) or both: TAp53 (DO-1, DAB) and p53β (KJC8, Fast Red). Scale bar represents 200 and 50 µm for 10× and 40× images, respectively. (**C**) Percentage of positive cells for each chromogen found in the cytoplasm, nucleus or both cellular compartments. (**D**) Percentage of positive cells for nuclear KJC8 and/or nuclear DO-1 and/or cytoplasmic DO-1. (**E**) Percentage of positive cells for cytoplasmic KJC8 and/or nuclear DO-1 and/or cytoplasmic DO-1; n: nuclear, c: cytoplasmic. Results are shown as individual values with the median represented by the black line.

**Figure 7 ijms-23-06670-f007:**
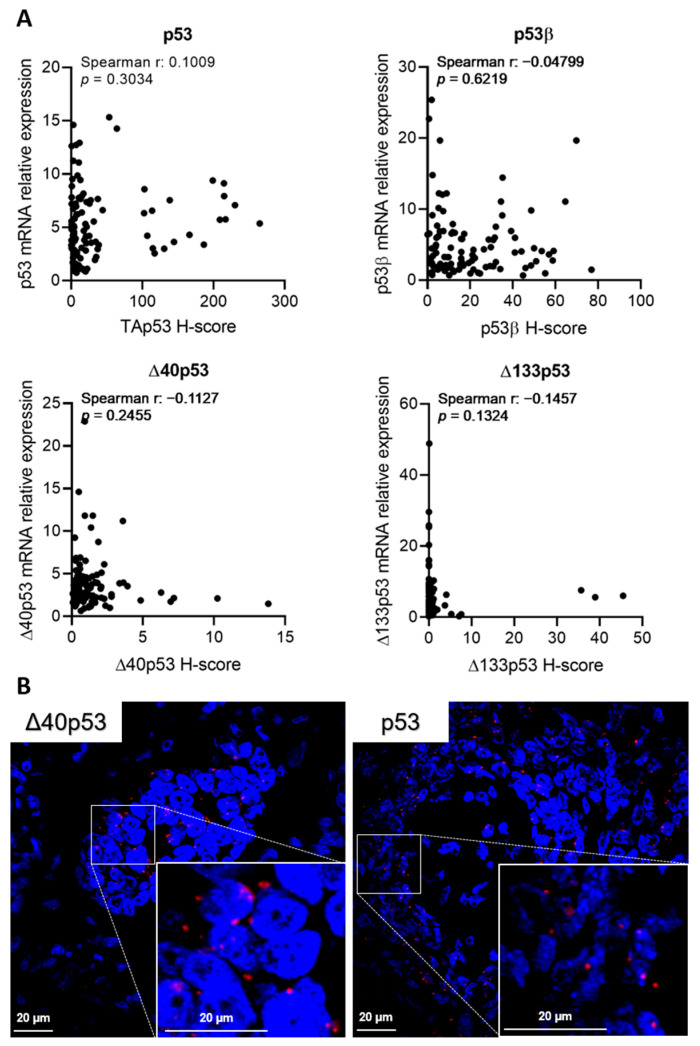
mRNA levels of p53 isoforms do not correlate with their protein expression levels. (**A**) Spearman rank correlation test was used to assess the correlation between mRNA data from our previous study [20] and protein levels (H-score) of p53, p53β, Δ40p53 and Δ133p53 (*n* = 108). (**B**) In situ hybridisation of a TNBC specimen for Δ*40p53* or *TP53* (shown in red). Cell nuclei were stained with DAPI. Scale bar represents 20 µm.

**Table 1 ijms-23-06670-t001:** Demographic data from 108 breast cancer cases used in this study.

Patient Information	
Age (Years)	
Average (±SD)	58.6 ± 15.1
Median	57.5
Range	28–90
<40	10
40–50	28
>50	70
Grade	
1	31 (28.1%)
2	24 (22.2%)
3	53 (49.1%)
Tumour size (mm)	
Average (±SD)	29.6 ± 15.8
Median	25
Range	7–90
≤25	55
>25	53
N^o^. of positive LNs	
0	50
1–3	40
>3	18
Hormone receptor status	
ER positive	80 (74.1%)
PR positive	73 (67.6%)
HER2 positive	18 (16.7%)
TNBC	15 (13.9%)

ER: oestrogen receptor; HER2: human epidermal growth factor receptor 2; LN: lymph nodes; PR: progesterone receptor; SD: standard deviation; TNBC: triple-negative breast cancer.

**Table 2 ijms-23-06670-t002:** Association of truncating *TP53* mutations with clinicopathological features.

Clinicopathological Features	Total	No Truncating *TP53* Mutation	Truncating *TP53* Mutation	Spearman r	*p* Value
	*n* = 103	*n* = 74 (72%)	*n* = 29 (28%)		
Age (years), median = 58				0.2281	0.0205 *
Range = 28–90					
<40	10	9	1		
40–50	27	21	6		
>50	66	44	22		
Grade				−0.8275	<0.0001 *
1 or 2	50	21	29		
3	53	53	0		
Tumour size (mm), median = 25				−0.2130	0.0307 *
≤25	53	35	18		
>25	50	39	11		
No. of positive LNs				−0.1270	0.2010
0	48	33	15		
1–3	37	25	12		
>3	18	16	2		
ER				0.3825	<0.0001 *
Positive	75	46	29		
Negative	28	28	0		
PR				0.3373	0.0005 *
Positive	70	43	27		
Negative	33	31	2		
HER2, (*n* = 102)				−0.2348	0.0176 *
Positive	18	17	1		
Negative	84	56	28		
TNBC				−0.2585	0.0084 *
Yes	15	15	0		
No	88	59	29		

Spearman’s correlation test was used to determine statistical significance. Results were considered significant at *p* < 0.05 (*). Correlation coefficient r: 0–0.19 is very weak, 0.2–0.39 is weak, 0.40–0.59 is moderate and 0.6–0.79 is strong. ER: oestrogen receptor; HER2: human epidermal growth factor receptor 2; LN: lymph nodes; PR: progesterone receptor; TNBC: triple-negative breast cancer.

**Table 3 ijms-23-06670-t003:** Association of high or low levels of p53 with clinicopathological features (*n* = 108).

Clinicopathological Features	Low Levels	High Levels	*p* Value
	*n* = 54	*n* = 54	
Age (years), median = 58			0.0361 *
Range = 28–90			
<40	9	1	
40–50	12	16	
>50	37	33	
Grade			0.8474
1 or 2	28	27	
3	26	27	
Tumour size (mm), median = 25			0.0039 *
≤25	20	35	
>25	34	19	
No. of positive LNs			0.3009
0	29	21	
1–3	17	23	
>3	8	10	
ER			0.1877
Positive	43	37	
Negative	11	17	
PR			0.5374
Positive	38	35	
Negative	16	19	
HER2, (*n* = 107)			0.8005
Positive	10	9	
Negative	44	45	
TNBC			0.0123 *
Yes	3	12	
No	51	42	

Chi-square was used to determine statistical significance. Results were considered significant at *p* < 0.05 (*). ER: oestrogen receptor; HER2: human epidermal growth factor receptor 2; LN: lymph nodes; PR: progesterone receptor; TNBC: triple-negative breast cancer.

## Data Availability

The data presented in this study are available on request from the corresponding author.

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
