# Peer review of "Cytoplasmic p53β Isoforms Are Associated with Worse Disease-Free Survival in Breast Cancer"

_ijms, 2022, doi:10.3390/ijms23126670_

Round 1

Reviewer 1 Report

This study analyzed p53 isoforms at the protein level in cellular and subcellular breast cancer tissues. High levels of cytoplasmic p53β isoforms, such as Δ133p53β, were significantly associated with worse disease-free survival in invasive ductal carcinomas. Furthermore, p53β could be a useful prognostic marker. 

The experiment was professionally designed and conducted, the data were rigorously analyzed and interpreted, and the results open new research leads. Based on these circumstances, I recommend the paper to be considered for publication. 

Reviewer 2 Report

In this study the authors analyze the role of p53β isoforms in the prognosis of breast cancer patients. Effectively, many evidences have recently emerged about the role of p53 isoforms in the prognosis of cancers and the topic is really interesting and deserves to be investigated. The authors provided convincing evidence for the association between p53β isoforms with worse disease free survival in breast cancer. 

Overall, I found the manuscript well written, I have some questions and suggestions that should be addressed.

-  The p53 isoforms are mainly localized in the nuclei. Some isoforms, as Δ40p53, have been shown actively upregulate the transcription of key oncogenes involved in cancer progression DOI: 10.1073/pnas.2103319118. May p53β to be involved in regulate some gene involved in cancer progression?  Could the authors investigate some genes potentially modulated by p53β  or at least discuss this point in the article?

- As the authors know mutant p53 has been widely observed  participate in the oncogenic process regulating a myriad of pathways, as autophagy, ROS, metabolism ect PMID: 30318520. Could p53β exerts on similar pathways that regulates also mutant p53, to lead worst prognosis? The authors should be more involved in functional characterization of p53β.. how is its role on autophagy, and cell proliferation of breast cancer cells?

- Also, in the tumor cells, exist simultaneously all isoforms or someone are more expressed? and what are the effect of these isoforms on wild type p53? are the dominant negative?

Round 2

Reviewer 2 Report

I thanks the authors for their detailed explanations made in the response letter. They provided satisfactory informations and addressed to all my concerns previously raised.

I really enojoyed read this paper and the authors comments, and I believe that it will serve as reference for further investigations in the field of p53 isoform biology.